# Comparison of microbiota in the cloaca, colon, and magnum of layer chicken

**Seo-Jin Lee, Seongwoo Cho, Tae-Min La, Hong-Jae Lee, Joong-Bok Lee, Seung-Yong Park, Chang-Seon Song, In-Soo Choi, Sang-Won Lee** \* 

College of Veterinary Medicine, Konkuk University, Seoul, Republic of Korea

\* odssey@konkuk.ac.kr

**Data Availability Statement:** All raw sequence reads files are available from the NCBI database (accession number: PRJNA604381).

**Funding:** This paper was supported by Konkuk University in 2016 to SWL.

## Abstract

Anatomically terminal parts of the urinary, reproductive, and digestive systems of birds all connect to the cloaca. As the feces drain through the cloaca in chickens, the cloacal bacteria were previously believed to represent those of the digestive system. To investigate similarities between the cloacal microbiota and the microbiota of the digestive and reproductive systems, microbiota inhabiting the colon, cloaca, and magnum, which is a portion of the chicken oviduct of 34-week-old, specific-pathogen-free hens were analyzed using a 16S rRNA metagenomic approach using the Ion torrent sequencer and the Qiime2 bioinformatics platform. Beta diversity via unweighted and weighted unifrac analyses revealed that the cloacal microbiota was significantly different from those in the colon and the magnum. Unweighted unifrac revealed that the cloacal microbiota was distal from the microbiota in the colon than from the microbiota in the magnum, whereas weighted unifrac revealed that the cloacal microbiota was located further away from the microbiota in the magnum than from the microbiota inhabiting the colon. *Pseudomonas* spp. were the most abundant in the cloaca, whereas *Lactobacillus* spp. and *Flavobacterium* spp. were the most abundant species in the colon and the magnum. The present results indicate that the cloaca contains a mixed population of bacteria, derived from the reproductive, urinary, and digestive systems, particularly in egg-laying hens. Therefore, sampling cloaca to study bacterial populations that inhabit the digestive system of chickens requires caution especially when applied to egg-laying hens. To further understand the physiological role of the microbiota in chicken cloaca, exploratory studies of the chicken's cloacal microbiota should be performed using chickens of different ages and types.

## Introduction

Avian gut microbiota displays certain features. First, avian gut microbiota aid in protecting host birds from pathogens and contribute to the development of the immune system of the hosts [1]. Second, antibiotics administered to these birds may affect the gut microbiota depending on the dose of the antibiotic used and the age of the birds [2]. Third, avian gut microbiota are saccharolytic rather than cellulolytic and help degrade polysaccharides

**Competing interests:** The authors have declared that no competing interests exist.

contained in poultry feed [3]. Finally, gut microbes may be affected by the body temperature of their avian host [4]. The most abundant bacterial genus in chicken gut varied depending on the type of sample and measuring techniques for bacterial population used in previous studies. Studies using gut contents showed that the most abundant bacterial genus in chicken gut was *Clostridium* [5–7]. The most abundant bacterial genus in chicken feces was *Bacteroides* in lean chickens, but *Clostridium* in fat chickens [8]. Another study showed that the most abundant bacterial genus in chicken feces was *Escherichia* except unclassified genus [9], while the other study showed that the most abundant bacterial genus in chicken feces was *Lactobacillus* [10]. A Study used cloacal swabs showed that the most abundant bacterial genus in cloaca of broilers was *Lactobacillus* [11]. Usually feces were collected to study the gut microbiota, because collecting feces is non-invasive. However, cloacal swab was preferred for collecting individual samples from birds. Recently, gut microbiota of juvenile ostriches was compared with those of feces and cloaca. In the study, cloacal microbiota was far different to microbiota in colon and feces [12, 13]. In contrast to this study, some of microbiota in cloaca of turkey were matched to microbiota in intestine in genus level [14]. These results raised the question of whether cloacal microbiota can represent the intestinal microbiota in chicken. Therefore, this study aimed to compare cloacal microbiota with those in colon and magnum, a part of oviduct in SPF laying hens.

## Materials and methods

### Sample collections

Eleven 34-week-old SPF laying chickens were used in this study. All experimental procedures were approved by the institutional animal care and use committee of Konkuk University (approval number KU17103-1). Cloacae were swabbed using the CLASSIQ swabs (Coppan, Murrieta, CA, USA), which were then suspended in 2 ml phosphate-buffered saline (PBS). The suspended samples were stored at -20˚C until DNA extraction for a day. Birds were euthanized using $CO_2$ gas and the magnum in the oviducts and colons were aseptically harvested. Mucosal area of the magnum and colon were scraped using the back of a scalpel and suspended in 1 ml of PBS and stored at -20˚C until DNA extraction for a day. Ten 30-week-old Hy-Line brown commercial layer chicken carcasses were used for the isolation of *Lactobacillus* spp. from the cloaca, colon, and magnum. Each location was swabbed with the CLASSIQ swab and the swab was streaked on De Man, Rogosa and Sharpe agar (MRS) agar. Streaked MRS agars were incubated in 37˚C for 48 h. Species of all grown colonies were identified via Matrix-assisted laser desorption/ionization and time-of-flight (MALDI-TOF) spectrometry and species of colonies not identified via MALDI-TOF were identified via 16S rRNA sequencing with 357F and 926R primers.

### Extraction of DNA and sequencing

Bacterial DNA was extracted in 1 ml of PBS using the DNeasy blood and tissue kit (Qiagen, Manchester, UK). Amplification of V2, V3, V4, V6-V7, and V9 regions of the 16S rRNA was conducted using primer sets from the Ion 16S Metagenomics kit (Thermo Fisher Scientific, Waltham, MA, USA). The Ion S5 XL sequencer and the Ion 530 chip were used for sequencing.

### Sequence analysis

A Qiime2 platform [15] was used for metagenome analysis via the Greengenes database (13_8 release) as the 16s rRNA gene reference [16]. The first 15 bases of all reads were removed, each sequence was truncated at position 150, and reads below the phred quality score 15 were filtered using DADA2 [17]. Chimeric sequences were detected via vsearch [18] and removed.

Operational taxonomic units (OTUs) were constructed with filtered sequences using a 99% identity option. The OTUs were classified with a Naive Bayes classifier [19]. Sampling depth was set up to 3000 feature counts for diversity metrics and alpha rarefaction. One magnum sample was excluded because it showed very different microbial components compared to the other magnum samples. Alpha diversity was measured using the Shannon index for non-phylogenetic alpha diversity metric [20]. Beta diversity was measured using unweighted unifrac [21] and weighted unifrac [22] for phylogenetic beta diversity. The Emperor tool was used to visualize principal coordinates analysis (PCoA) plots [23]. To evaluate associations among microbiota in the cloaca, colon and magnum, the pairwise permutational multivariate analysis of variance (PERMANOVA) statistic was used and p-values were produced with 999 permutation tests. Relative frequencies of taxa for each group were displayed in bar plots. Differentially abundant taxa of each group were identified via analysis of microbiome composition (ANCOM) [24]. A SourceTracker2 [25] was used to calculate the contribution of microbiota in the colon and magnum to microbiota in the cloaca.

## Results

### Sequencing results

The cloaca, colon, and magnum samples of 11 SPF hens were analyzed. Subsequently, 6,707,244 raw reads (mean 209,601.375 ± 88,595.49) were obtained (Table 1). Following filtering, 1,315,288 reads (mean 41,102.75 ± 27,937) were obtained and classified into 1192 OTUs, which clustered at a 99% identity level. The raw sequence reads were deposited in the NCBI sequence read archive under BioProject accession number: PRJNA604381.

### Alpha diversity and beta diversity analysis

Alpha diversity of microbiota in the cloaca, colon, and magnum of 11 SPF hens were analyzed via the Shannon index, which is used to measure the non-phylogenetic alpha diversity metric. The Shannon index of microbiota in the cloaca was lower than those in the colon and magnum (Fig 1).

However, this difference was not significant as indicated by the pairwise Kruskal–Wallis test for the Shannon index (Table 2).

Beta-diversity analysis using an unweighted unifrac metric was performed to analyze distance among the microbiota in the cloaca, colon, and magnum. Microbiota in the cloaca, colon, and magnum were grouped separately on the PCoA plot (Fig 2).

In the pairwise PERMANOVA, the cloaca, colon, and magnum showed statistically significant differences in microbial composition, furthermore the microbiota in the cloaca and colon were farther apart than the microbiota in the cloaca and the magnum (Table 3).

Beta-diversity analysis using a weighted unifrac metric was also performed to analyze distance among the microbiota in the cloaca, colon, and magnum. Microbiota in the cloaca, colon, and magnum were grouped separately on the PCoA plot (Fig 3).

Pairwise PERMANOVA showed that the cloaca, colon, and magnum showed statistically significant differences in microbial composition, furthermore the microbiota in the cloaca and magnum were farther apart than the microbiota in the cloaca and colon (Table 4).

### Taxonomic analysis

The relative taxa abundance plots at the genus level show the 20 most abundant taxa in the three groups. The most abundant genus in the cloaca was *Pseudomonas*, followed by *Gallibacterium*, *Lactobacillus*, *Bacteroides*, and unclassified *Actinomycetales*. The most abundant genus

**Table 1. Raw reads, filtered reads, and total OTUs of each sample.**

| Samples | Raw reads | filtered reads | OTUs |
|---|---|---|---|
| Cloaca1 | 218949 | 27012 | 203 |
| Cloaca2 | 214261 | 29918 | 146 |
| Cloaca3 | 262902 | 37777 | 152 |
| Cloaca4 | 258339 | 30567 | 154 |
| Cloaca5 | 252877 | 34276 | 170 |
| Cloaca6 | 303497 | 37132 | 98 |
| Cloaca7 | 340755 | 37701 | 111 |
| Cloaca8 | 434301 | 70203 | 207 |
| Cloaca9 | 208477 | 21132 | 127 |
| Cloaca10 | 253007 | 27500 | 201 |
| Cloaca11 | 209453 | 22595 | 148 |
| Colon1 | 230704 | 8928 | 143 |
| Colon2 | 190807 | 9963 | 154 |
| Colon3 | 151946 | 5281 | 103 |
| Colon4 | 149177 | 6690 | 120 |
| Colon5 | 185545 | 3502 | 81 |
| Colon6 | 172814 | 8161 | 139 |
| Colon7 | 195808 | 6609 | 126 |
| Colon8 | 98102 | 3474 | 83 |
| Colon9 | 175641 | 8161 | 141 |
| Colon10 | 184051 | 7398 | 97 |
| Colon11 | 212088 | 8556 | 125 |
| Magnum1 | 110363 | 7933 | 107 |
| Magnum2 | 68544 | 6684 | 335 |
| Magnum3 | 106573 | 7876 | 123 |
| Magnum4 | 60056 | 7039 | 204 |
| Magnum5 | 84874 | 11503 | 132 |
| Magnum6 | 315157 | 22100 | 188 |
| Magnum7 | 431246 | 34927 | 235 |
| Magnum8 | 181004 | 29282 | 343 |
| Magnum9 | 193660 | 24836 | 183 |
| Magnum11 | 252266 | 27712 | 243 |

in the colon was *Lactobacillus*, followed by *Bacteroides*, unclassified *Bacteroidales*, unclassified *Lachnospiraceae*, and *Faecalibacterium*. The most abundant genus in the magnum was *Flavobacterium*, followed by *Lactobacillus*, unclassified *Moraxellaceae*, *Pseudomonas*, and *Megamonas*. To perform a taxonomic analysis of the shared microbiota in the cloaca, colon, and magnum, a sample each was pooled from one group respectively. Relative common taxa abundance plots at the genus level show the 10 most abundant taxa in the 3 groups (Fig 4). *Lactobacillus* spp. was the most abundant common taxa among each group.

The most abundant common genus in the cloaca was *Pseudomonas*, followed by *Lactobacillus*, unclassified *Burkholderiales*, *Megamonas*, and unclassified *Lachnospiraceae*. The most abundant common genus in the colon was *Lactobacillus*, followed by *Bacteroides*, *Faecalibacterium*, unclassified *Bacteroidales*, and unclassified *Lachnospiraceae*. The most abundant common genus in the magnum was *Lactobacillus*, followed by *Pseudomonas*, *Megamonas*, unclassified *Lachnospiraceae*, and *Faecalibacterium*. The most abundant common genus

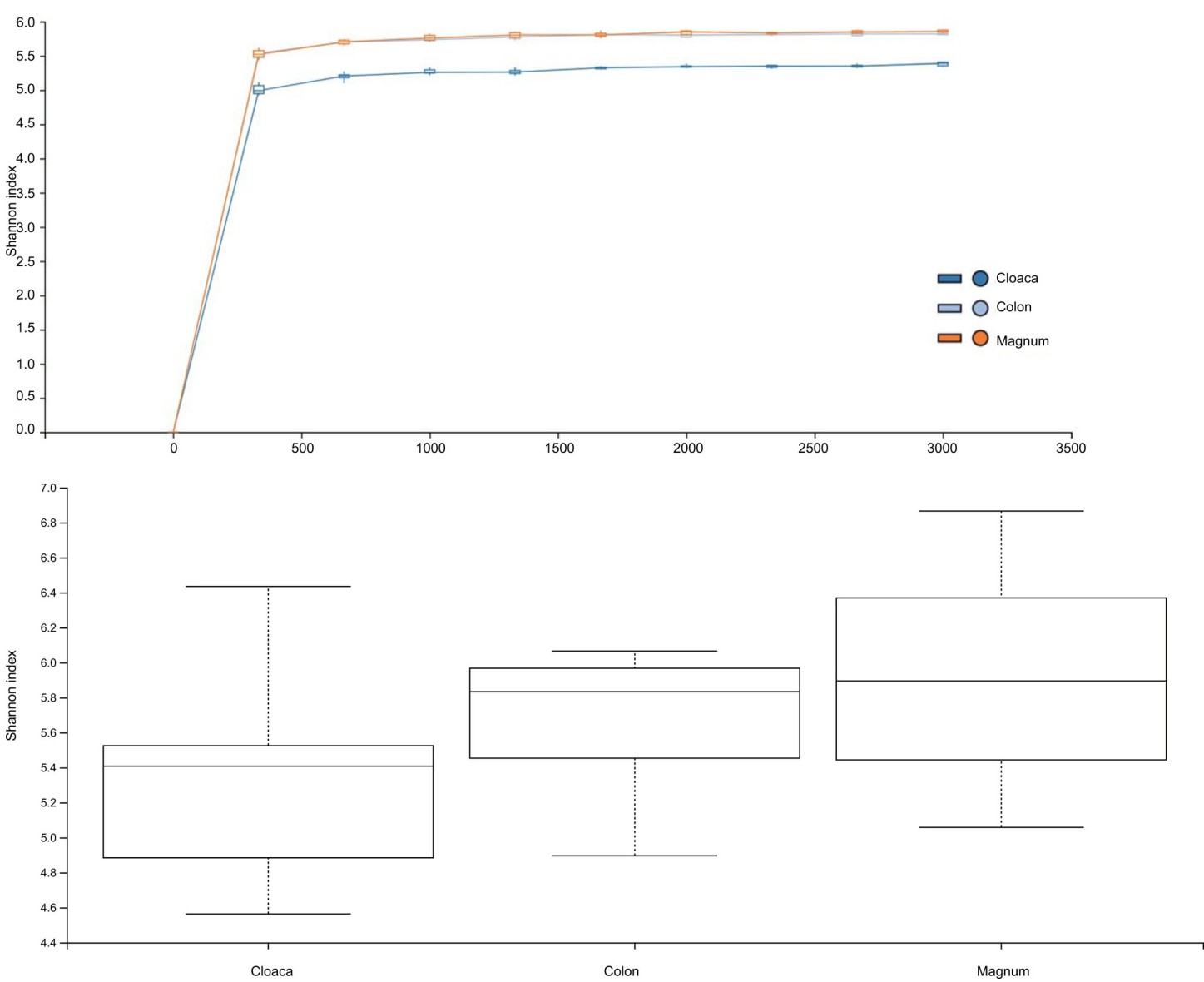

**Fig 1. Comparison of the Shannon index between the cloaca, colon, and magnum.** Microbiota in the cloaca, colon, and magnum of SPF laying hens were analyzed via Shannon's index. (A) Rarefaction curve for Shannon's index. The dark blue line represents the cloaca, the orange line represents the magnum, and the light (sky) blue line represents the colon. (B) Shannon's index for each group. Box plots show the quartiles, median, and extremities of the values.

among all groups was *Lactobacillus*, followed by *Pseudomonas*, *Megamonas*, *Bacteroides*, and unclassified *Lachnospiraceae*. There were 5 core taxa in the cloaca, 15 core taxa in the colon, and 20 core taxa in the magnum (Table 5).

**Table 2. Pairwise Kruskal-Wallis tests for Shannon's index of each group.**

| Group 1 | Group 2 | H | p-value | q-value |
|---------|---------|---|---------|---------|
| Cloaca | Colon | 2.588214 | 0.107662 | 0.161492 |
| Cloaca | Magnum | 4.462810 | 0.034640 | 0.103921 |
| Colon | Magnum | 0.714050 | 0.398103 | 0.398103 |

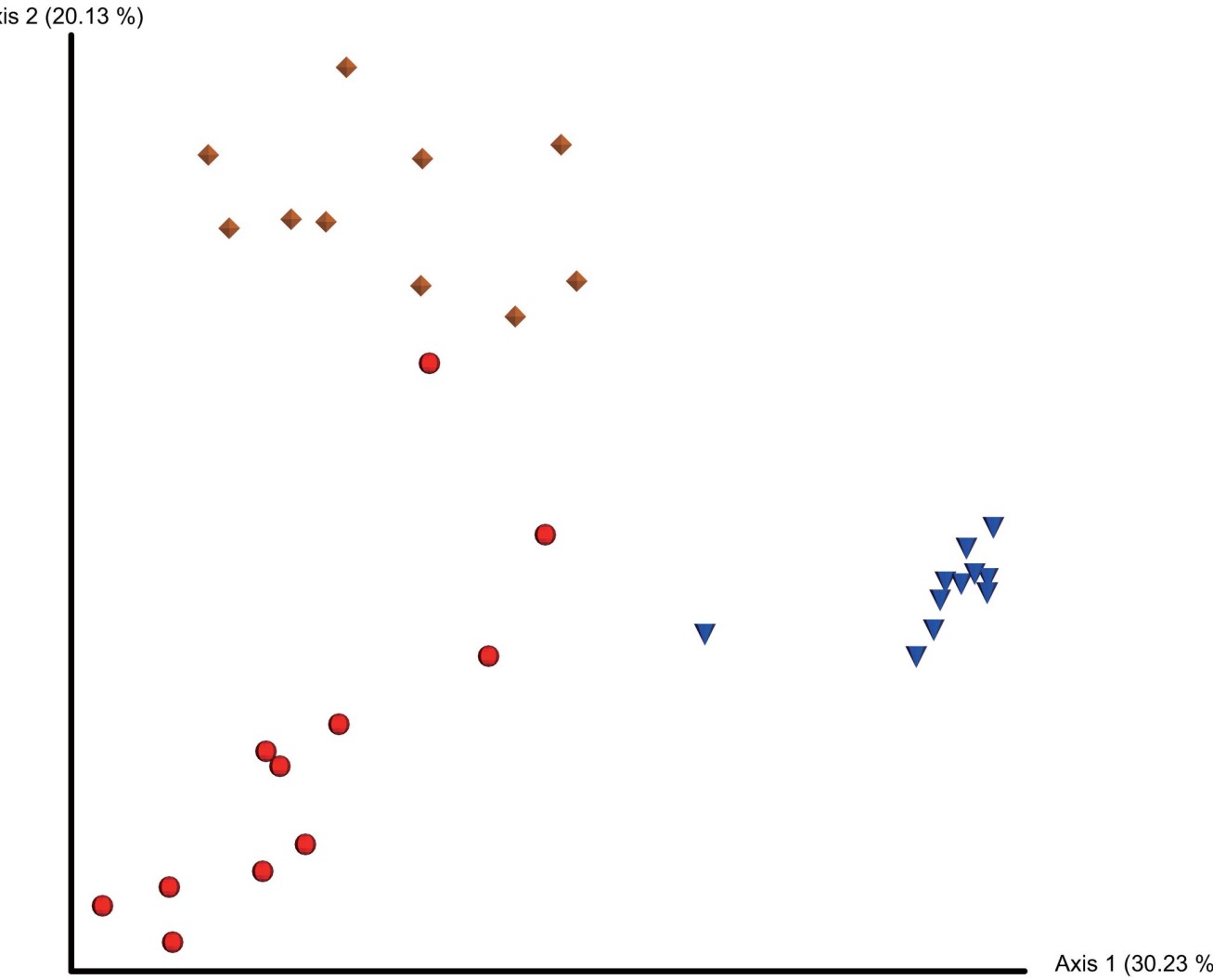

**Fig 2. PCoA plot based on unweighted unifrac distance matrix.** PCoA plots demonstrating unweighted unifrac distance among microbiota in the cloaca, colon, and magnum of laying hens. Red spheres represent the cloaca, blue spheres represent the colon, and yellow diamonds represent the magnum.

## Detection of *Lactobacillus* spp. at each location

*Lactobacillus* spp. was the most common genus among each group. However, since the sequencing results of metagenomic analysis using 16S rRNA amplicon usually are not accurate enough to determine the correct bacterial species, we could not say the detected *Lactobacilli* were the same species or not. Therefore, additionally the dominant species of *Lactobacillus* spp. inhabiting each sampling site were investigated using culture technique. *Lactobacillus* spp.

**Table 3. Pairwise PERMANOVA results based on unweighted unifrac distance matrix.**

| Group 1 | Group 2 | pseudo-F | p-value | q-value |
|---------|---------|----------|---------|---------|
| Cloaca | Colon | 15.239907 | 0.001 | 0.001 |
| Cloaca | Magnum | 7.236330 | 0.001 | 0.001 |
| Colon | Magnum | 13.728121 | 0.001 | 0.001 |

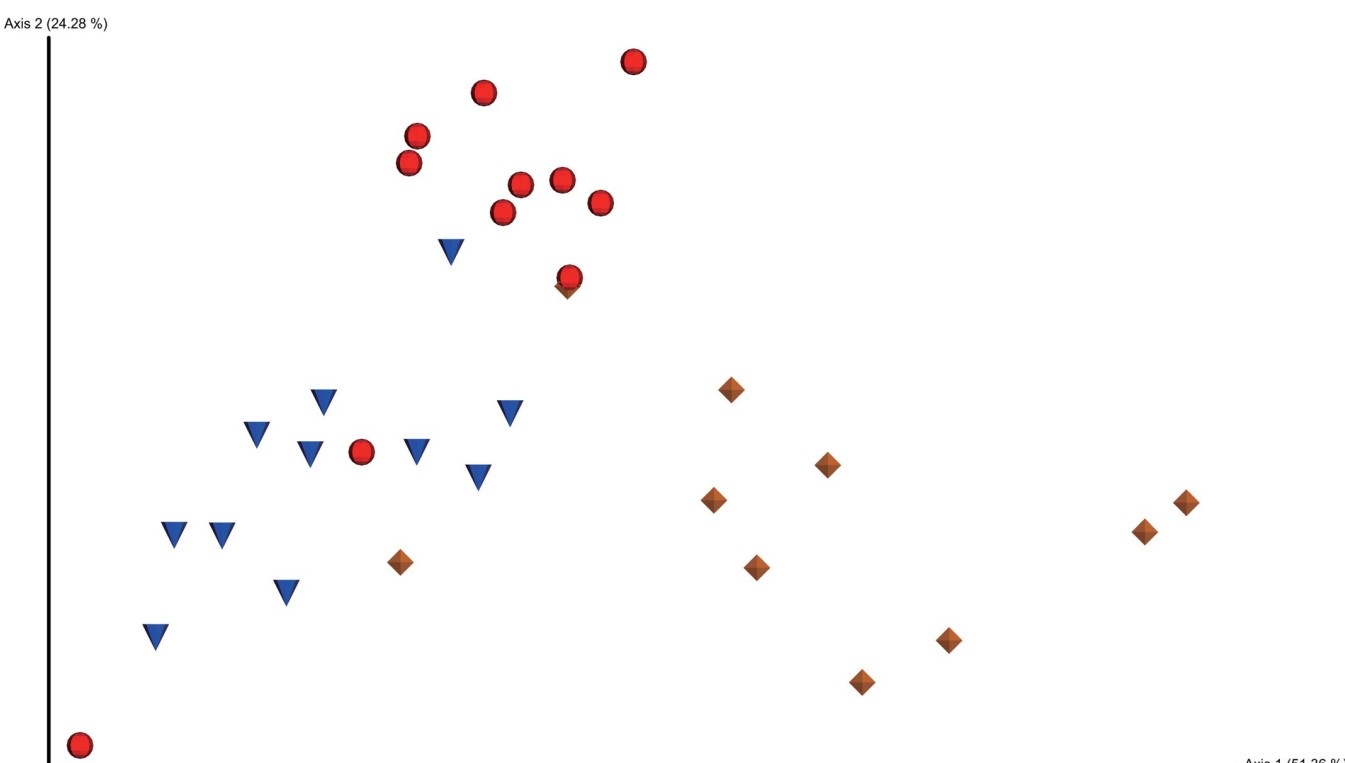

**Fig 3. PCoA plot based on weighted unifrac distance matrix.** PCoA plots demonstrating weighted unifrac distance among microbiota in the cloaca, colon, and magnum of laying hens. Red spheres represent the cloaca, blue spheres represent the colon, and yellow diamonds represent the magnum.

from each location were identified via MALDI-TOF spectrometry and 16s rRNA sequencing. Eleven *Lactobacillus* spp. were detected in the cloaca, 5 in the colon, and 5 in the magnum. *Lactobacillus reuteri* was the most dominant *Lactobacillus* sp. in the cloaca and colon, and *Lactobacillus vaginalis* was the most dominant *Lactobacillus* sp. in the magnum (Fig 5).

## Differential abundance analysis

ANCOM was used to identify differentially abundant genera among the cloaca, colon and magnum. *Gallibacterium*, *Enterococcus*, *Janthinobacterium*, unclassified *Gammaproteobacteria*, *Actinomyces*, *Helococcus*, unclassified *Pasteurellaceae*, *Stenotrophomonas*, *Morganella*, and *Comamonas* were differentially abundant in cloaca. Unclassified *Actinomycetales*, unclassified *Enterobacteriaceae*, *Acinetobacter*, unclassified *Xanthomonadaceae*, and *Corynebacterium* were differentially abundant in the cloaca and the magnum compared with the colon. *Flavobacterium*, unclassified *Rhodobacteraceae*, *Brevundimonas*, unclassified *Microbacteriaceae*, unclassified *Caulobacteraceae*, unclassified *Flavobacteriaceae*, *Propionibacterium*, *Methylobacterium*, and *Rhodobacter* were differentially abundant in the magnum. Unclassified *RF39*,

**Table 4. Pairwise PERMANOVA results based on weighted unifrac distance matrix.**

| Group 1 | Group 2 | pseudo-F | p-value | q-value |
|---------|---------|----------|---------|---------|
| Cloaca | Colon | 8.492881 | 0.003 | 0.0030 |
| Cloaca | Magnum | 10.851457 | 0.001 | 0.0015 |
| Colon | Magnum | 17.966760 | 0.001 | 0.0015 |

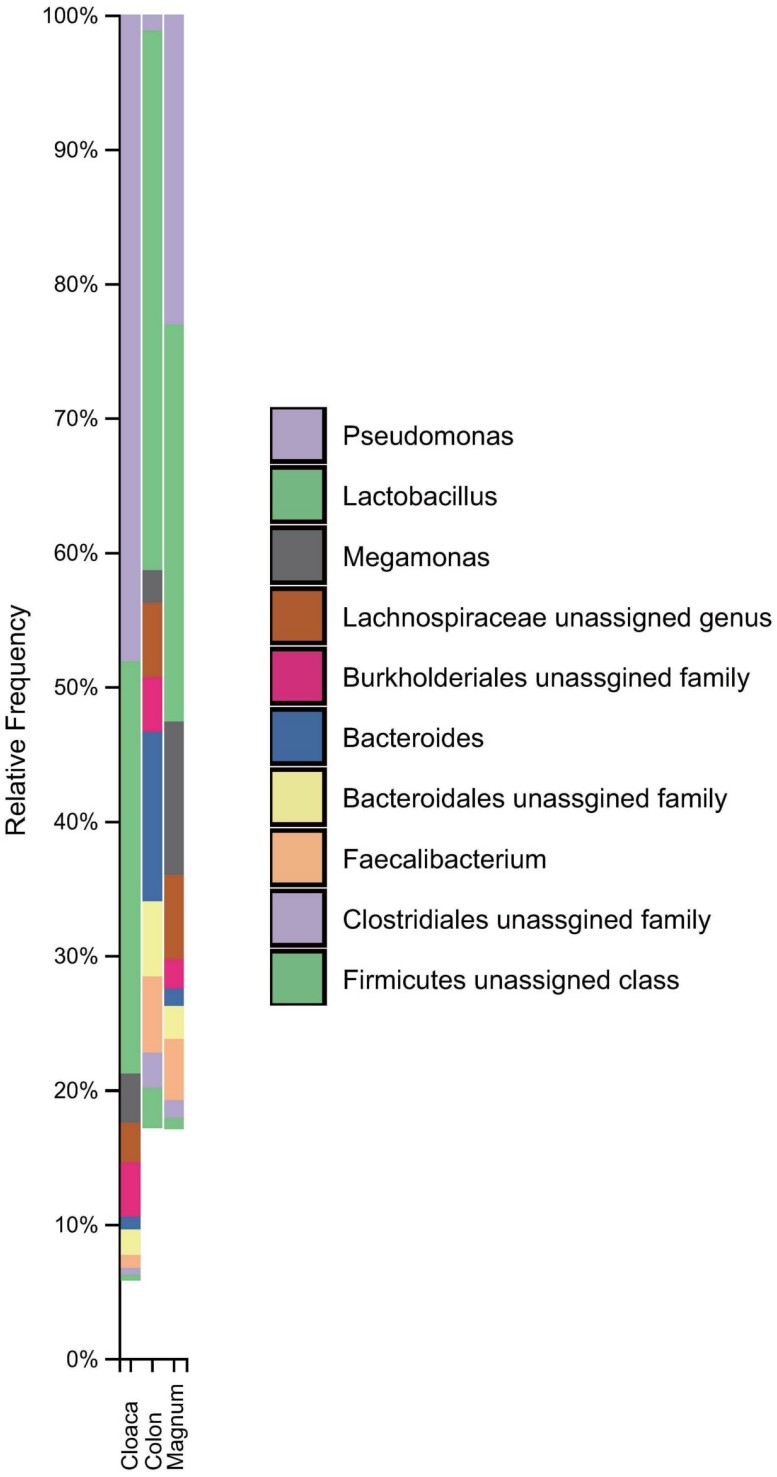

**Fig 4. Relative frequency of ten of the most abundant common taxa among all groups at the genus level.** Ten of the most abundant taxa, classified by different colors, are shown. Each bar indicates the relative frequencies of ten of the most abundant common taxa among all groups at genus level.

**Table 5. Core taxa\* of each sampling group.**

| Group | Taxa |
|---|---|
| Cloaca | *Actinomyces* |
| | *Enterococcus* |
| | *Lactobacillus* |
| | Unclassified *Actinomycetales* |
| | Unclassified *Gammaproteobacteria* |
| Colon | *Bacteroides* |
| | *Coprobacillus* |
| | *Lactobacillus* |
| | *Megamonas* |
| | Unclassified *Firmicutes* |
| | Unclassified *Bacteroidales* |
| | Unclassified *Burkholderiales* |
| | Unclassified *Clostridiales* |
| | Unclassified *RF39* |
| | Unclassified *Coriobacteriaceae* |
| | Unclassified *Lachnospiraceae* |
| | Unclassified *Rikenellaceae* |
| | Unclassified *Ruminococcaceae* |
| | Unclassified *Veillonellaceae* |
| Magnum | *Bacteroides* |
| | *Brevundimonas* |
| | *Faecalibacterium* |
| | *Flavobacterium* |
| | *Lactobacillus* |
| | *Megamonas* |
| | *Methylobacterium* |
| | *Pseudomonas* |
| | *Rhodobacter* |
| | Unclassified *Betaproteobacteria* |
| | Unclassified *Actinomycetales* |
| | Unclassified *Bacteroidales* |
| | Unclassified *Burkholderiales* |
| | Unclassified *Clostridiales* |
| | Unclassified *Caulobacteraceae* |
| | Unclassified *Enterobacteriaceae* |
| | Unclassified *Lachnospiraceae* |
| | Unclassified *Microbacteriaceae* |
| | Unclassified *Moraxellaceae* |
| | Unclassified *Ruminococcaceae* |
| | Unclassified *Xanthomonadaceae* |

\* Genera detected in all samples in each group were considered as core genera.

unclassified *Coriobacteriaceae*, and unclassified *Bacteroidales* were differentially abundant in the colon (S1 Table).

At the genus level, 56 genera were common to the cloaca, colon, and magnum (Fig 6).

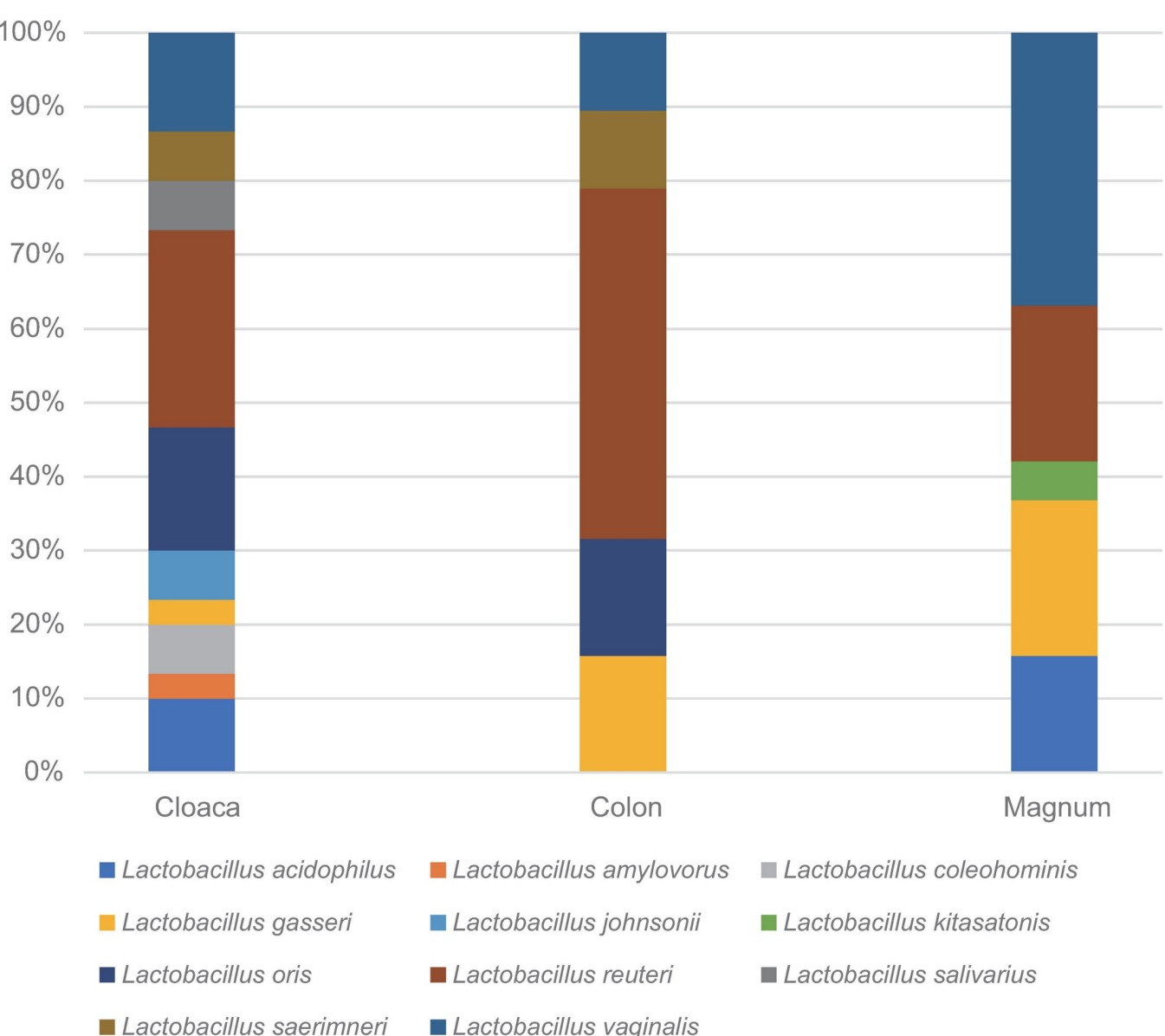

**Fig 5. The distribution of *Lactobacillus* spp. detected at each location.** Detected *Lactobacillus* spp. at each location are indicated with different colors. Each bar indicates the relative detected frequencies of *Lactobacillus* spp. among all groups.

### Origin of microbiota in chicken cloaca

The SourceTracker 2 was used to analyze the origin of the microbiota in the cloaca and each sample from one group was pooled. When the cloaca was assigned as the sink, 0.0669 of microbiota in the colon and 0.0809 of microbiota in the magnum contributed to the microbiota in the cloaca, whereas the highest contribution (0.8714) to the microbiota in the cloaca was from an unknown source (Table 6).

## Discussion

With the development of sequencing technology, research on gut microbiota is becoming active, and new roles of microorganisms in the intestine have been revealed [26]. Using a

B

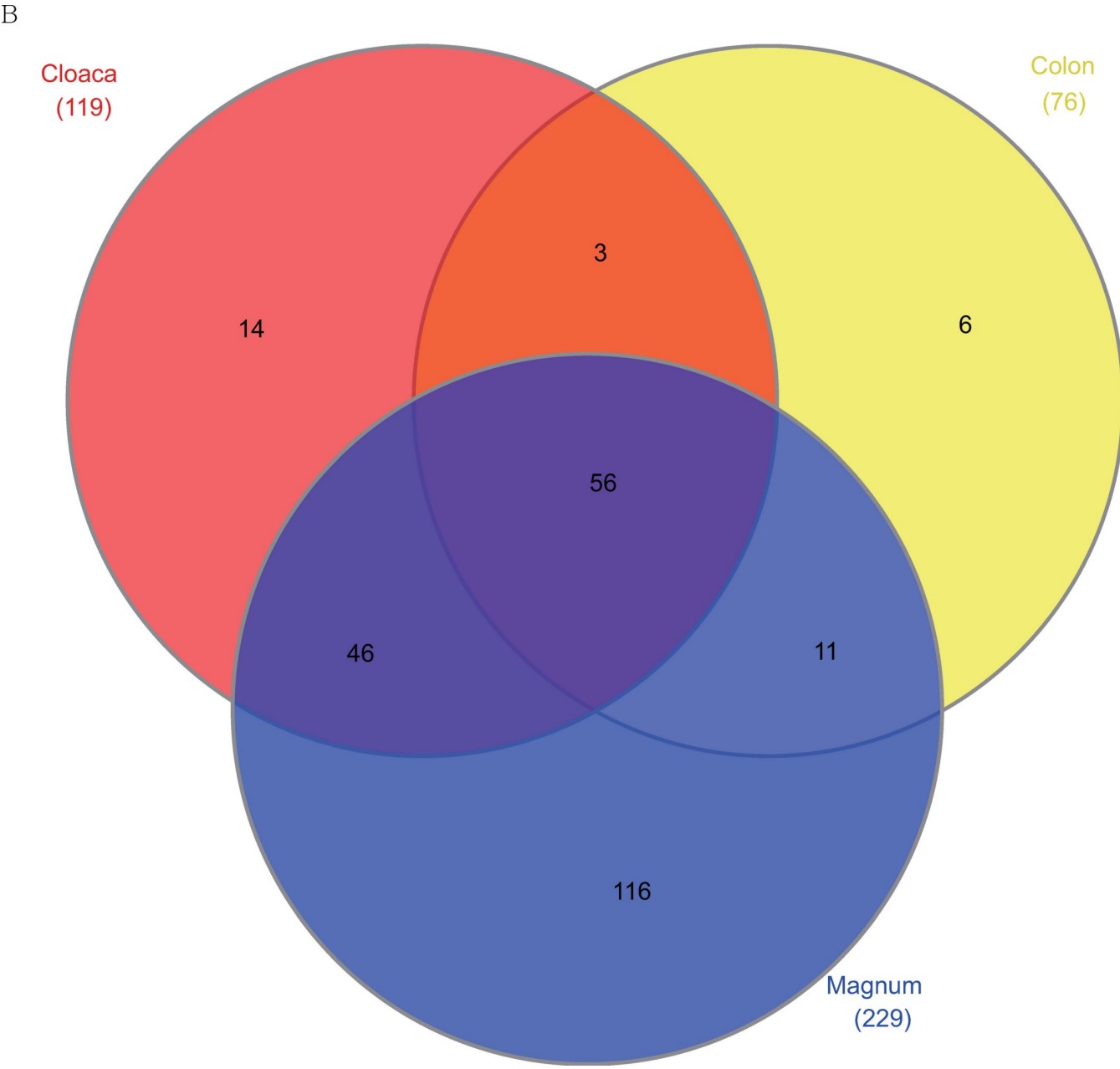

**Fig 6. Common and unique phylotypes at the genus level among each group.** Venn diagram demonstrating the number of common or unique phylotypes at the genus level among the groups. Phylotypes observed in each part were counted.

**Table 6. Contribution of each source to each sink.**

| Sink | Colon | Magnum | Cloaca | Unknown |
|---|---|---|---|---|
| Colon | - | 0.1029(0.0171) | 0.0257(0.007) | 0.8714(0.0176) |
| Magnum | 0.0192(0.0074) | - | 0.0111(0.0054) | 0.9697(0.0093) |
| Cloaca | 0.0669(0.0138) | 0.0809(0.0089) | - | 0.8522(0.0161) |

* Standard deviations are in parentheses.

suitable sample for the study of gut microbiota is a very important factor in obtaining valuable results. Cloacal swab is a non-invasive and multiple sampling method for the same individual for the study of poultry intestinal microbiota [13]. Anatomically, cloaca is connected to the end of the digestive system, however in case of a hen, it also connects to the urinary and reproductive systems [13], so there was a question of whether the microbiota of cloaca can represent gut microbiota. In this study, we compared and analyzed microbiota present in the colon, oviduct, and cloaca of laying hens to assess whether it is possible to study the intestinal microbiota of laying hens using cloacal swabs. The results of this study indicated that the cloacal microbiota was significantly different from those in the colon and the magnum in the beta diversity analysis. Since colon and magnum samples were taken with scalpel and cloaca samples with swab, there may be a possibility that the microbiota may be different due to the difference in sampling method. Results of beta diversity analysis were slightly different between unweighted unifrac and weighted unifrac. Unweighted unifrac is a qualitative measure that does not consider the relative abundance of taxa, whereas weighted unifrac is a quantitative measure that considers the relative abundance of taxa [22]. In relative taxa abundance, the most abundant common genus in the cloaca was *Pseudomonas*, while the most abundant common genus in the colon and magnum was *Lactobacillus*. The cloaca is more aerobic than the colon and the magnum [27], and *Pseudomonas* is an aerobic bacteria [28] that may easily colonize the cloaca compared to the colon and the magnum. The most abundant common genus among all different sites was *Lactobacillus*. We used SPF white leghorn chickens to perform 16S rRNA metagenome analysis, while the Hy-Line brown commercial chickens were used in order to culture *Lactobacillus spp.* in all sampling sites. Although it is possible that different *Lactobacillus spp.* present in different breeds of chicken, culture results were consistent with those of the 16S rRNA metagenome analysis as all sampling sties contained *Lactobacillus spp. Lactobacillus reuteri* was the most dominant *Lactobacillus* spp. in the cloaca and colon, while *Lactobacillus vaginalis* was the most dominant *Lactobacillus* spp. in the magnum. *Lactobacillus reuteri* is an inhabitant in gastrointestinal tract in mammal and bird. Administration of *Lactobacillus reuteri* could improve growth of chickens having avian growth depression [29] and protect chickens from *Salmonella* Enteritidis challenge infection [30]. Unfortunately, role of *Lactobacillus vaginalis* in chicken has never been studied before. *Lactobacillus gasseri* were observed in magnum and colon in this study. *Lactobacillus gasseri* has been reported that it can produce lactocillin [31] and bacteriocin which have antimicrobial activity [32]. A small number of *Lactobacillus* spp. abundance have been linked to the development of bacterial vaginosis in human [33, 34]. According to our previous research [35], very few *Lactobacillus* spp. were present in the oviduct of unmatured pullets. Laying hen's oviduct can be more easily infected by external bacteria than unmatured pullets, which may be one of the reasons that *Lactobacilli* increase in the oviduct of laying hens. Probably in the oviduct of chicken, *Lactobacilli* can protect the host against pathogenic bacterial infections. Since different *Lactobacillus* spp. were present in the intestine and oviduct of laying hens, there is a possibility that variety *Lactobacillus* spp. may protect the host from different species of bacterial pathogens in different body sites. Cloacal *Lactobacillus* spp. probably formed by the mixed population of *Lactobacilli* derived from the magnum and colon, and some *Lactobacillus* spp., which were absent in both of the magnum and colon. It can be assumed that cloacal lactobacilli are derived from not only the magnum and colon but also an unknown source (i.e., the environment). When the SourceTracker2 was used to find the original sources of the cloacal microbiota, the highest contribution (0.8714) was from an unknown source. Thus, in summation, although the colon and magnum contributed some species to the cloaca, overall, the microorganisms originating from the colon and the magnum were few. In conclusion, microbiota in the cloaca do not represent the microbiota in the digestive tract in egg laying chicken. Most notably, the SourceTracker2

results showed that the cloacal microbiota largely came from an unknown source, which is most likely an outside source from the ambient aerobic environment rather than from the digestive or reproductive track. Therefore, sampling cloaca to study bacterial populations that inhabit the digestive system of chickens requires caution especially when applied to egg-laying hens. To further understand the physiological role of the microbiota in chicken cloaca, exploratory studies of the chicken's cloacal microbiota should be performed using chickens of different ages and types.

## Supporting information

**S1 Table. Percentile abundance between groups.**
(XLSX)

## Author Contributions

**Conceptualization:** Joong-Bok Lee, Seung-Yong Park, Chang-Seon Song, In-Soo Choi, Sang-Won Lee.

**Formal analysis:** Seo-Jin Lee.

**Funding acquisition:** Sang-Won Lee.

**Investigation:** Seo-Jin Lee.

**Methodology:** Seo-Jin Lee, Seongwoo Cho, Tae-Min La, Hong-Jae Lee.

**Resources:** Seo-Jin Lee, Seongwoo Cho, Tae-Min La, Hong-Jae Lee.

**Supervision:** Sang-Won Lee.

**Writing – original draft:** Seo-Jin Lee.

**Writing – review & editing:** Joong-Bok Lee, Seung-Yong Park, Chang-Seon Song, In-Soo Choi, Sang-Won Lee.

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
