## [Decision Letter · Decision Letter 0]

19 May 2020

PONE-D-20-06110

Comparison of microbiota in the cloaca, colon, and magnum of layer chicken

PLOS ONE

Dear Dr Lee,

Thank you for submitting your manuscript to PLOS ONE. After careful consideration, we feel that it has merit but does not fully meet PLOS ONE’s publication criteria as it currently stands. Therefore, we invite you to submit a revised version of the manuscript that addresses the points raised during the review process.

I feel that the manuscript is dealing with a good topic but lacks in the quality of preparation. I agree with reviewers in particular, I would like to inform that I care about criticism of referee#3 more and please review the referee comments and make your peer revision.

We would appreciate receiving your revised manuscript by Jul 03 2020 11:59PM. To enhance the reproducibility of your results, we recommend that if applicable you deposit your laboratory protocols in protocols.io, where a protocol can be assigned its own identifier (DOI) such that it can be cited independently in the future. For instructions see: http://journals.plos.org/plosone/s/submission-guidelines#loc-laboratory-protocols

We look forward to receiving your revised manuscript.

Kind regards,

Arda Yildirim, Ph.D.

Academic Editor

PLOS ONE

Additional Editor Comments (if provided):

The study is well presented, I feel that the manuscript is dealing with a good topic but lacks in the quality of preparation. The main problem found in the manuscript is related to the some aspects of the methodology, poor discussion and typo errors or ambiguous phrases or sentences. It is necessary to improve the manuscript by examining the questions that need to be clarified in a way. Please be aware of the manuscript should be presented according to guidelines for authors of Plos One. For your guidance, you can check the reviewers' comments.

2. Thank you for stating the following financial disclosure: "no"

Reviewers' comments:

Reviewer's Responses to Questions

**Comments to the Author**

1. Is the manuscript technically sound, and do the data support the conclusions?

Reviewer #1: Yes

Reviewer #2: Yes

Reviewer #3: No

2. Has the statistical analysis been performed appropriately and rigorously? 

Reviewer #1: Yes

Reviewer #2: Yes

Reviewer #3: I Don't Know

3. Have the authors made all data underlying the findings in their manuscript fully available?

Reviewer #1: Yes

Reviewer #2: Yes

Reviewer #3: Yes

4. Is the manuscript presented in an intelligible fashion and written in standard English?

Reviewer #1: Yes

Reviewer #2: Yes

Reviewer #3: Yes

5. Review Comments to the Author

Reviewer #1: The manuscript is presented in an intelligible fashion. It is written well and clear. The data provided, along with the figures and it's discussion support the conclusions in this manuscript. The authors have made all data underlying the findings in their manuscript fully available.

Reviewer #2: In terms of methodology, the work is correct. Although the purpose of the study, according to the authors, is clearly defined, i.e. „ To investigate similarities between the cloacal

microbiota and the microbiota of the reproductive and digestive systems, microbiota

inhabiting the colon, cloaca, and magnum of 34-week-old, specific-pathogen-free

(SPF) hens were analyzed….” , The authors did not clearly explain the scientific or application significance of the analyzes performed. This is another qualitative and quantitative analysis of the microbiota of selected sections of the gastrointestinal tract of hens of a specific age group. The novelty of the conducted research should be indicated in relation to the available data or the results of other authors known so far. The study designed in this way, and above all the way of discussing the results obtained in the discussion, significantly reduces the value of the manuscript.

Abstract:

line 15-16-I suggest rewriting this sentence

Line 18-22: The authors did not analyze reproductive system microbiota in comparative studies, so the purpose of the research is not entirely consistent with the analysis. Obviously, comparing the biota of the cloaca and the other two sections of the intestine, we can indirectly conclude but still only indirectly, so it would be necessary to verify the presentation of the goal

Introduction:

line 44: please explain what "products" mean

Line55-56: Since the cloaca is a combination of the few systems, by definition the composition of the microbiota from the material taken from the cloaca cannot be directly related to the microbiota of the digestive tract.

Materials and methods:

line 66: in line with the assumptions of written work, each abbreviation should be explained at least once

Line 66 and 70: please specify the maximum storage time

Line 62, 70, 109, 119 and others: how many samples have been tested? In one place Authors give 10 in another 11 chickens - please verify

Results

Line 205: It is interesting why the authors focused on Lactobacillus species analysis. This needs more clarification.

Discussion

Unfortunately, the discussion is mostly a repetition of the results. According to the discussion, the results obtained by the Authors here, were already confirmed by other Authors, therefore my question remains open: What is the novelty of the research? In the conclusion, the Authors only drew attention to the fact that the results obtained from cloaca (microbiota analysis) should be applied with great caution to the analysis of gastrointestinal biota - this was already known, so it is difficult to find new elements in the presented studies. Perhaps if the discussion were supplemented with a discussion of the importance of individual taxa shown in the study in relation to the impact on the physiological processes of birds in individual sections of the gastrointestinal tract would be much more valuable. In its present form, the manuscript is another study in the form of registering the presence / absence of specific taxa. In addition, the Authors did a more accurate species analysis of Lactobacillus but did not use these results to enrich the discussion in suitable way.

Reviewer #3: The purpose of the study is to determine whether in birds, the microflora in the cloaca is a good estimate of the microflora in the gut. The cloaca in birds serves as the only opening for the digestive, reproductive, and urinary tracts, all of which may have their own microbiome. If the microflora of the cloaca is a mixture of these three different organ systems, it may not be particularly representative of the digestive system. The study samples the microflora of 11 specific-pathogen-free (SPF) chickens in three different regions: the cloaca, the colon (digestive system), and the magnum (part of the oviduct, which is part of the reproductive system). The authors investigated the microflora by amplicon sequencing of the 16S rRNA gene. The authors found that the microflora differs between these three different regions using a number of community ecology statistics. They found that bacteria belonging to the Genus Lactobacillus were among the most common. To determine the community of Lactobacillus in greater detail, the authors used another 10 Korean commercial layer chicken carcasses and cultured Lactobacillus bacteria on MRS agar followed by species identification via MALDI-TOF spectrometry and 16S rRNA sequencing. This study found that different Lactobacillus species were dominant in the different regions.

One problem with the study is that methodological differences are often confounded with the main factors of interest. First example, the cloaca is sampled using a swab whereas the magnum and colon were sampled using a scalpel. So differences in the microflora between the cloaca and the other two regions (magnum and colon) are potentially confounded by the method of sampling the bacteria. Second example, the authors use two different types of chickens for the two different parts of their study: SPF chickens to determine the general bacterial community and Korean commercial layer chickens to determine the community of Lactobacillus bacteria. Different strains of chickens could have very different microflora, but this obvious possibility is not discussed. Third example, the microflora in the first part of the study was amplified via PCR whereas the microflora in the second part of the study was amplified via culturing on MRS agar. These different methods of microflora amplification (PCR versus culture) will change the microflora, which makes inference difficult. Fourth example, the authors used 16S rRNA gene sequencing in the first study and MALDI-TOF spectrometry in the second study to identify bacterial species. These different identification methods could also influence the composition of the bacterial community that was detected. Nowhere in the manuscript do the authors acknowledge or discuss how the use of all of these different methods could bias their results.

In summary, there are many differences between the first and second experiment: breed of chicken, method of microflora amplification (PCR versus culture), and bacterial identification method (16S rRNA sequencing verus MALDI-TOF spectrometry). All these differences undermine the rationale for combining these two experiments into a single study. PLOS ONE requires that study has a sound experimental design, which I do not find to be the case for this study. I believe that this manuscript would be better suited to a more specialized journal on poultry science.

6. PLOS authors have the option to publish the peer review history of their article (what does this mean?). If published, this will include your full peer review and any attached files.

Reviewer #1: No

Reviewer #2: No

Reviewer #3: No

---

## [Author Response · Author response to Decision Letter 0]

1 Jul 2020

Reviewer #2

Reviewer #2: In terms of methodology, the work is correct. Although the purpose of the study, according to the authors, is clearly defined, i.e. „ To investigate similarities between the cloacal

microbiota and the microbiota of the reproductive and digestive systems, microbiota

inhabiting the colon, cloaca, and magnum of 34-week-old, specific-pathogen-free

(SPF) hens were analyzed….” , The authors did not clearly explain the scientific or application significance of the analyzes performed. This is another qualitative and quantitative analysis of the microbiota of selected sections of the gastrointestinal tract of hens of a specific age group. The novelty of the conducted research should be indicated in relation to the available data or the results of other authors known so far. The study designed in this way, and above all the way of discussing the results obtained in the discussion, significantly reduces the value of the manuscript.

- Following reviewer’s suggestion, we rewrote the introduction section to clarify the aim of this study and revised the discussion section with reviewing of other previous studies. 

Abstract:

line 15-16-I suggest rewriting this sentence

-We rewrote the sentence in line 15-16 on page 2

Line 18-22: The authors did not analyze reproductive system microbiota in comparative studies, so the purpose of the research is not entirely consistent with the analysis. Obviously, comparing the biota of the cloaca and the other two sections of the intestine, we can indirectly conclude but still only indirectly, so it would be necessary to verify the presentation of the goal-? 

-We added a detailed description of the magnum in line 20 on page 2

Introduction:

line 44: please explain what "products" mean 

-We rewrote the sentence to make clear in line 49-50 on page 4

Line55-56: Since the cloaca is a combination of the few systems, by definition the composition of the microbiota from the material taken from the cloaca cannot be directly related to the microbiota of the digestive tract. 

-We added that results of the previous studies which compared microbiota in the digestive tract and cloaca swab in line 55-61 on page 4. In addition, we revised the aim of this study in line 61-64 on page 4-5 

Materials and methods:

line 66: in line with the assumptions of written work, each abbreviation should be explained at least once

-We added explanation of the abbreviation in line 72 and 79 on page 5

Line 66 and 70: please specify the maximum storage time 

-We added the maximum storage time in line 76 on page 5

Line 62, 70, 109, 119 and others: how many samples have been tested? In one place Authors give 10 in another 11 chickens - please verify- 

-We used 11 SPF chickens for analyzing microbiota in each sampling site and used 10 Hy-Line brown commercial layer chickens to culture Lactobacillus spp. in each sampling site.

Results

Line 205: It is interesting why the authors focused on Lactobacillus species analysis. This needs more clarification.-

-We explained why we focused on Lactobacillus species in line 212-217 on page 13 

Discussion

Unfortunately, the discussion is mostly a repetition of the results. According to the discussion, the results obtained by the Authors here, were already confirmed by other Authors, therefore my question remains open: What is the novelty of the research? 

In the conclusion, the Authors only drew attention to the fact that the results obtained from cloaca (microbiota analysis) should be applied with great caution to the analysis of gastrointestinal biota - this was already known, so it is difficult to find new elements in the presented studies. 

-We added the novelty and significance of this research in line 258-270 on page 15 and line 314-316 on page 17

Perhaps if the discussion were supplemented with a discussion of the importance of individual taxa shown in the study in relation to the impact on the physiological processes of birds in individual sections of the gastrointestinal tract would be much more valuable.

-We added physiological effect of dominant Lactobacillus species in line 287-293 on page 17 

 In its present form, the manuscript is another study in the form of registering the presence / absence of specific taxa. In addition, the Authors did a more accurate species analysis of Lactobacillus but did not use these results to enrich the discussion in suitable way.

-We added the discussion in line 295-302 on page 16

Reviewer #3

Reviewer #3: The purpose of the study is to determine whether in birds, the microflora in the cloaca is a good estimate of the microflora in the gut. The cloaca in birds serves as the only opening for the digestive, reproductive, and urinary tracts, all of which may have their own microbiome. If the microflora of the cloaca is a mixture of these three different organ systems, it may not be particularly representative of the digestive system. The study samples the microflora of 11 specific-pathogen-free (SPF) chickens in three different regions: the cloaca, the colon (digestive system), and the magnum (part of the oviduct, which is part of the reproductive system). The authors investigated the microflora by amplicon sequencing of the 16S rRNA gene. The authors found that the microflora differs between these three different regions using a number of community ecology statistics. They found that bacteria belonging to the Genus Lactobacillus were among the most common. To determine the community of Lactobacillus in greater detail, the authors used another 10 Korean commercial layer chicken carcasses and cultured Lactobacillus bacteria on MRS agar followed by species identification via MALDI-TOF spectrometry and 16S rRNA sequencing. This study found that different Lactobacillus species were dominant in the different regions. One problem with the study is that methodological differences are often confounded with the main factors of interest. 

First example, the cloaca is sampled using a swab whereas the magnum and colon were sampled using a scalpel. So differences in the microflora between the cloaca and the other two regions (magnum and colon) are potentially confounded by the method of sampling the bacteria. 

-Many previous studies measured microbiota in reproductive system, digestive tract using a scalpel and microbiota in cloaca using cotton swab. Therefore, we followed the same method with precedent studies. We added the possibility that various sampling methods can affect the study results in line 270-272 on page 15 

Second example, the authors use two different types of chickens for the two different parts of their study: SPF chickens to determine the general bacterial community and Korean commercial layer chickens to determine the community of Lactobacillus bacteria. Different strains of chickens could have very different microflora, but this obvious possibility is not discussed. 

-We added the controversy in discussion in line 281-286 on page 16

Third example, the microflora in the first part of the study was amplified via PCR whereas the microflora in the second part of the study was amplified via culturing on MRS agar. These different methods of microflora amplification (PCR versus culture) will change the microflora, which makes inference difficult. 

-We explained why we should use culture method to determine exact species of Lactobacillus in all sampling sites. We added the sentence in the result section in line 212-217 on page 13

Fourth example, the authors used 16S rRNA gene sequencing in the first study and MALDI-TOF spectrometry in the second study to identify bacterial species. These different identification methods could also influence the composition of the bacterial community that was detected. Nowhere in the manuscript do the authors acknowledge or discuss how the use of all of these different methods could bias their results.

-The results which were deduced by MALDI-TOF spectrometry were consistent with those of the 16S rRNA metagenome sequencing in genus level. We added the sentence in discussion in line 281-286 on page 16

In summary, there are many differences between the first and second experiment: breed of chicken, method of microflora amplification (PCR versus culture), and bacterial identification method (16S rRNA sequencing verus MALDI-TOF spectrometry). All these differences undermine the rationale for combining these two experiments into a single study. PLOS ONE requires that study has a sound experimental design, which I do not find to be the case for this study. I believe that this manuscript would be better suited to a more specialized journal on poultry science.

---

## [Decision Letter · Decision Letter 1]

21 Jul 2020

Comparison of microbiota in the cloaca, colon, and magnum of layer chicken

PONE-D-20-06110R1

Dear Dr. Lee,

We’re pleased to inform you that your manuscript has been judged scientifically suitable for publication and will be formally accepted for publication once it meets all outstanding technical requirements.

Kind regards,

Arda Yildirim, Ph.D.

Academic Editor

PLOS ONE

Additional Editor Comments (optional):

Please make a small correction:

Lines 304-306: please correct and unify spelling „cloacal lactobacilli”

Reviewers' comments:

Reviewer's Responses to Questions

**Comments to the Author**

1. If the authors have adequately addressed your comments raised in a previous round of review and you feel that this manuscript is now acceptable for publication, you may indicate that here to bypass the “Comments to the Author” section, enter your conflict of interest statement in the “Confidential to Editor” section, and submit your "Accept" recommendation.

Reviewer #2: All comments have been addressed

2. Is the manuscript technically sound, and do the data support the conclusions?

Reviewer #2: Yes

3. Has the statistical analysis been performed appropriately and rigorously? 

Reviewer #2: N/A

4. Have the authors made all data underlying the findings in their manuscript fully available?

Reviewer #2: Yes

5. Is the manuscript presented in an intelligible fashion and written in standard English?

Reviewer #2: Yes

6. Review Comments to the Author

Reviewer #2: The manuscript has improved significantly, and the authors have referred to all comments, therefore I recommend manuscript for publication.

Please make a small correction:

Lines 304-306: please correct and unify spelling „cloacal lactobacilli”

7. PLOS authors have the option to publish the peer review history of their article (what does this mean?). If published, this will include your full peer review and any attached files.

Reviewer #2: No

---

## [Editor Report · Acceptance letter]

24 Jul 2020

PONE-D-20-06110R1 

Comparison of microbiota in the cloaca, colon, and magnum of layer chicken 

Dear Dr. Lee:

I'm pleased to inform you that your manuscript has been deemed suitable for publication in PLOS ONE. Congratulations! Your manuscript is now with our production department. 

Kind regards, 

on behalf of

Dr. Arda Yildirim 

Academic Editor

PLOS ONE